# Efficient Recursive Fréchet Mean Estimation

## Abstract

Estimating the mean is a key aspect of statistical analysis. Doing such an estimation on Riemannian manifolds is complex due to lacking a closed form solution. The gradient descent algorithm is commonly used to approximate the Fréchet mean across various applications. Although generally effective, it can be problematic when the dataset is large as each computation of the gradient can be costly or when the mean is not uniquely defined as in positively curved manifolds. This paper introduces a tree-based, recursive Fréchet mean estimator (RFME), designed for data on the hypersphere. We prove the weak consistency of the RFME with the true mean and demonstrate its computational efficiency and accuracy through two simulations and two real-world case studies. We compare our algorithm to the standard gradient descent approach and to the incremental Fréchet mean estimator (iFME), a state of the art algorithm that efficiently estimates the mean. Lastly, our algorithm is a generalization of the iFME and thus our algorithm has more flexibility.

## 1 Introduction

In statistical inference, the center of a dataset is often of interest as it is a fundamental measure of the location of the data. The mean is a cornerstone of many statistical methods such as regression, principle geodesic analysis, and convex analysis. Under the scheme of Euclidean spaces, the sample mean is canonically defined as $\bar{x}_n := \sum_{i=1}^{n} x_i / n$. This definition is consistent with other terms such as barycentres, centroid, and expectation (in probability measures). This definition, however, assumes a linear space and well-defined addition and multiplication operations which are not always reasonable.

With advancements in measurement techniques, manifold-valued data has become increasingly common in many research fields. Unlike data in a vector spaces (e.g., Euclidean space, $\mathbb{R}^n$), manifold-valued data typically comes from spaces without an additive structure, meaning that the traditional method of calculating the mean by dividing the sum by the number of observations is no longer applicable. Given a space equipped with a metric, the distance between points can be determined. The Fréchet mean, which identifies the mean as the point that minimizes the total squared-distance to all observations, provides an alternative approach. The common way for estimating the Fréchet mean (FM) is through the gradient descent method, presented by Karcher (1977) and Pennec (1999). Le (2001) and Le (2004) then demonstrated the applicability of this method on complete, simply connected manifolds.

Given the computational time and storage usage requirements of the gradient descent method, many studies have proposed incremental mean estimation techniques to address this issue. In Euclidean space, the mean can computed in an iterative manner as

$$\bar{x}_n = \frac{n-1}{n}\bar{x}_{n-1} + \frac{1}{n}x_n. \tag{1}$$

This method has been generalized to manifolds with non-linear structure. Sturm (2003) first introduced an inductive mean estimator, and proved its existence and uniqueness, on the spaces of non-positive definite curvature (NPC). Later, Arnaudon et al. (2012) presented a stochastic version of the algorithm, and Lim & Pálfia (2014b) extended the method as the weighted inductive mean on the cone of positive-definite Hermitian matrices. Properties of the inductive mean has also been analyzed on specific manifolds such as the hypersphere (Salehian et al. (2015)), Symmetric Positive-Definite (SPD) matrices equipped with the affine invariant Riemannian metric (Ho

et al. (2013)), SPD matrices equipped with the Stein metric (Salehian et al. (2013a)), CAT(0) spaces (Feragen et al. (2011)), Grassmannian manifolds (Chakraborty & Vemuri (2015)), and the Stiefel manifold (Chakraborty & Vemuri (2019)). The convergence of the inductive mean has been well-established on manifolds due to its stochastic nature. Lim & Pálfia (2014a) and Holbrook (2012) also provided deterministic proofs in terms of the practical application. With the inductive mean estimator, a variety of algorithms can be extended to learning with manifold-valued data, such as Convolution Neural Networks (Chakraborty et al. (2022)), Recurrent Neural Networks (Chakraborty et al. (2018)), nonlinear regression (Banerjee et al. (2016)), and movement primitive learning (Daab et al. (2023)).

In this paper, we introduce the Recursive Fréchet mean estimator (RFME), a tree-based inductive algorithm to approximate the mean of a dataset. This is akin to constructing a binary tree where the leaves are the data, intermediate parents are the means of their children, and thus the root of the tree represents the mean estimator of the entire dataset. This can be viewed as a generalization of incremental Fréchet mean estimator (iFME, Salehian et al., 2015), and the estimator is computed in pairs of the point at each step, rather than following an order. We show that RFME is a consistent estimator of the FM and that it is more computationally efficient than the gradient descent approach. Additionally, our simulations on synthetic data and real-world cases suggest our approach is comparable to iFME in terms of time consumption and convergence rate, with additional benefit of being parallelizable.

The rest of the paper is organized as follows. Section 2 covers the preliminaries of Riemannian manifolds, geodesics, and inductive means, which are foundational for our presented algorithm. Section 3 provides our algorithm and the theoretical guarantee for the convergence of RFME to the true Fréchet mean. Section 4 presents experiments on both synthetic and real data, and Section 5 concludes the paper.

## 2 BACKGROUND

### 2.1 RIEMANNIAN MANIFOLDS

Here we present a brief introduction into Riemannian manifolds and the necessary backgrounds and notations. For a more thorough handling of manifolds and differential geometry we refer to Absil et al. (2008), Do Carmo (1992), and Lee (2018).

Let $\mathcal{M}$ denote a $d$-dimensional complete, smooth Riemannian manifold. At each point $p \in \mathcal{M}$ we have an associated *tangent space*, $T_p\mathcal{M}$. Each tangent space is a vector space and consists of all elements tangent to the manifold $\mathcal{M}$ at $p$. The collection of all tangent spaces is referred to as the tangent bundle, $T\mathcal{M} = \{T_p\mathcal{M} | p \in \mathcal{M}\}$. Each tangent space has an associated inner product, $\langle \cdot, \cdot \rangle_p$, and the collection of all such inner products is the Riemannian metric tensor, $\{\langle \cdot, \cdot \rangle_p | p \in \mathcal{M}\}$. This inner product, in turn, allows us to consider geometric properties of the manifold such as angles and lengths. The length $\mathcal{L}(\cdot)$ of a path $\alpha : [0, 1] \to \mathcal{M}$ is $\mathcal{L}(\alpha) = \int_0^1 \|\dot{\alpha}(t)\|_{\alpha(t)}^{1/2} dt$, where $\| \cdot \|_p^2 = \langle \cdot, \cdot \rangle_p$ and $\dot{\alpha}(t) = \frac{d}{dt}\alpha(t)$. The distance between two points $p, q$ on the manifold is the length of the shortest path between them and denoted $d(p, q) := \inf_\alpha \mathcal{L}(\alpha)$. The path of minimal length is referred to as a *geodesic*.

Geodesics are not always unique, e.g. the antipodal point on a sphere, however all such geodesics have the same length. The set of such points, where the uniqueness fails, is known as the cut locus. Given a geodesic $\alpha$ starting at $p$ and initial velocity $\dot{\alpha}(0) = v$, the exponential map at $p$, denoted as $\exp_p : T_p\mathcal{M} \to \mathcal{M}$, is defined by $\exp_p v = \alpha(1)$. The map is a diffeomorphism in a sufficiently small neighborhood of $p$, bounded by the cut locus. Within this neighborhood, the inverse of exponential map, called the inverse exponential map or log map, is given by $\log_p q = v$, where $q = \alpha(1)$. The Riemannian distance then can also be given by $d(p, q) = \|\log_p q\|_p = \|\log_q p\|_q$.

### 2.2 FRÉCHET AND KARCHER MEAN

Arguably, the most fundamental statistic of a dataset is the average or mean. On a manifold we do not use the typical definition of the mean $\bar{x} = 1/n \sum x_i$ as there is no guarantee this estimate will

lie on the manifold. Further, the addition and multiplication operations are not well-defined. This estimate is hence referred to as the *extrinsic* mean. The classical *intrinsic* extension of this definition is the Fréchet mean defined as

$$\mu = \operatorname{argmin}_{x \in \mathcal{M}} \frac{1}{2} \sum d^2(x, x_i).$$

The Fréchet mean is thus the point on the manifold which minimizes the sum of square distances from the data (i.e., the variance) (Fréchet, 1948). This is sometimes referred to as the Karcher mean after Hermann Karcher who did extensive theoretical work on its convergence (Karcher, 1977). Estimating this mean is typically done using a gradient descent approach. Letting $E(x) = \frac{1}{2} \sum_{i=1}^{n} d^2(x, x_i)$ be the variance energy, the gradient is $\nabla_x E(x) = -\sum_{i=1}^{n} \log_x x_i$ where $\log$ is the log map. We then update the estimate with exponential map as $\hat{x} = \exp_x(-\eta \nabla_x E(x))$ where $\eta$ is a tuning parameter, the step size, and proceed to the next iteration. This optimization requires computing $n$ many log maps per iteration which can be costly; for instance, on the space of symmetric positive-definite matrices with the affine Riemannian metric, the log map requires matrix inversion which is inherently costly. Note that some refer to the set of local minimizers of the variance function as the *Karcher Means*, we assume we have a unique global optimizer.

## 2.3 Incremental Fréchet Mean Estimator

Computing the empirical Fréchet mean using gradient descent has a time complexity of $O(nk)$, with sample size $n$ and number of iterations $k$. This can be computationally demanding, as it requires processing the entire dataset during each iteration. To address this issue, particularly in the context of streaming and incoming data, Salehian et al. (2013b) proposed the following generalized form of an incremental algorithm:

$$\mu_1 = x_1$$
$$\mu_k = \operatorname{argmin}_{x \in \mathcal{M}} (\omega_k d^2(\mu_{k-1}, x) + (1 - \omega_k) d^2(x_k, x)),$$

where $\mu_{k-1}$ is the $(k-1)$th mean estimator with weight $\omega_k$. The weight $\omega_k$ allows a weighted average to be computed given the importance of the datapoint $x_k$. Here $\mu_n$ can be interpreted as an estimate of the empirical FM of $n$ datapoints. This estimation is consistent but sacrifices accuracy for small sample sizes while benefiting from not needing to recompute the FM every time new data is introduced. Further, for one to recompute the FM the server needs to store the entire dataset, so this algorithm alleviates such a burden.

In Euclidean space, the mean of two points always lies along the line segment connecting them; the analogous idea on Riemannian manifolds is that the mean of two points always lies along the geodesic between them. Moving along a geodesic from an estimate of the mean in the direction of a new datapoint is the idea behind the incremental Fréchet mean estimator (iFME, Salehian et al., 2015). It is, thus, useful to characterize a general midpoint of a geodesic. Let $\alpha$ be a geodesic from $p \in \mathcal{M}$ to $q \in \mathcal{M}$, we further denote $\alpha_t(p, q) = \alpha(t)(p, q)$, where $t \in [0, 1]$.

The iFME is an algorithm to efficiently estimate the mean for streaming data and can be formulated for an equally weighted dataset as follows:

$$\hat{\mu}_1 = x_1$$
$$\hat{\mu}_k = \alpha_{\frac{1}{k}}(\hat{\mu}_{k-1}, x_k).$$

This is a generalization of equation 1 to Riemannian manifolds. The estimator updates along the geodesic, that is, for each iteration, $\hat{\mu}_{k-1}$ moves $1/k$ of the distance along the geodesic connecting $\hat{\mu}_{k-1}$ to the new data point $x_k$. Theoretical results on non-positive curvature manifolds (NPC, Sturm (2003)) and on the sphere (Salehian et al. (2015)) provide guarantees that such a inductive estimator is both consistent and efficient on certain manifolds.

## 3 Recursive Mean Algorithm

Suppose we have a dataset $D \subset \mathcal{M}$ with $D = \{x_1, x_2, \ldots, x_n\}$. We aim to combine elements of the dataset in a binary sort of fashion, thus it makes sense to consider $n = 2^k + 1 + r$ where $0 \le r < 2^k$. This notation is typical in number theory where $r$ refers to the remainder.

We organize the dataset as the leaves of a binary tree, with each internal node representing the weighted mean of its two children. Then the recursive relationship is defined as

$$\hat{\mu}_{a,b}^{(\omega_1+\omega_2)} = \alpha_\tau(\hat{\mu}_{a+1,2b-1}^{(\omega_1)}, \hat{\mu}_{a+1,2b}^{(\omega_2)})$$

with $\tau = \frac{\omega_2}{\omega_1+\omega_2}$ being a weighted midpoint on the geodesic. The indices may be a bit cumbersome, but the first subscript index refers to the layer from bottom to top, the second index is the element index of that level. Generally, we have $a \in \{0, 1, 2, \ldots, k+1\}$, $k+2$ layers in total, and $\max(b) \leq 2^a$ for each layer. The superscript is a weight with the leaves having unit weight (unless specify).

We wish to consider all the data so the total weight of our estimate will be $n$, hence we have our recursive Fréchet mean estimator (RFME) to be $\hat{\mu}_{0,1}^{(n)}$. In our setting, the bottom layers is the data, $\hat{\mu}_{k+1,i}^{(1)} = x_i$ for $i \in \{1, \ldots, n\}$, and we arrange them from left to right, so that for each nodes with two children, the left one always comes from a complete binary tree. Typically, if r is even, then $\hat{\mu}_{k,\max(b)}^{(1)} = \hat{\mu}_{k+1,n}^{(1)} = x_n$, that is, we keep the single data to the parent layer. This rule applies to any layer with odd number of nodes. A detailed description of the algorithm can be found in Algorithm 1.

This organization is a generalization of iFME as we can set the tree to have $n + 1$ many layers, the bottom layer being $x_1$ and each subsequent layer including one datapoint. As opposed to the iFME, this algorithm can take advantage of parallel programming as, at any layer, each node is independent of every other node.

Figure 1a illustrates an example of the recursive estimation process. In this case $n = 2^k$, and we have that every $\tau = \frac{1}{2}$ which greatly simplifies the notation. Figure 1b compares the empirical Fréchet mean estimator using gradient descent method, incremental estimator and recursive estimator. The samples are generated uniformly on the hemisphere of $S^2$, as discussed in 4.1.

---

**Algorithm 1** Recursive Fréchet mean estimator

1: Given a set of data points $\{x_i\}_{i=1}^n$ with corresponding weights $\{w_i\}_{i=1}^n$ (default 1).
2: Assign $\hat{\mu}_{t,i}^{(w_i)} = x_i$, where $t = \lceil \log_2 n \rceil$.
3: Compute $\hat{\mu}_{t-1,j}^{(w_j')} = \alpha_\tau(\hat{\mu}_{t,2j-1}^{(w_{2j-1})}, \hat{\mu}_{t,2j}^{(w_{2j})})$, where $\tau = \frac{w_{2j}}{w_{2j+1}+w_{2j}}$ and $w_j' = w_{2j} + w_{2j+1}$, for $j = 1, \ldots, \lfloor \frac{M}{2} \rfloor$, $M = |\{\hat{\mu}_{t,i}^{(w_i)}\}|$.
4: If $M$ is odd, $\hat{\mu}_{t-1,\lfloor \frac{M}{2} \rfloor+1}^{(w_{\lfloor \frac{M}{2} \rfloor+1}')} = \hat{\mu}_{t,M}^{(w_M)}$.
5: If $t \neq 0$, set $t = t - 1$, and return to step 3.
6: Output $\hat{\mu}_{0,1}^{(\sum_{i=1}^n w_i)}$.

---

### 3.1 THEORETICAL GUARANTEES

Here we show that $\hat{\mu}_{0,1}^{(n)}$, as defined earlier, is an unbiased and consistent estimator of $\mathbb{E}[x]$. Let us first consider the hypersphere. Since the recursive mean is defined on the geodesic, we want to convert it into a probability space. Let $\eta$ be any point on the manifold, and denote $\tilde{\mu}_1 = \log_\eta(\hat{\mu}_{a+1,2b-1}^{(\omega_1)})$ and $\tilde{\mu}_2 = \log_\eta(\hat{\mu}_{a+1,2b}^{(\omega_1)})$. Here the logarithm (or log) is the mapping from the manifold $\mathcal{M}$ to the tangent spaces of $\eta$, thus $\tilde{\mu}_1, \tilde{\mu}_2 \in T_\eta \mathcal{M}$. Therefore, we can get the midpoint on the tangent space, as it is a vector space, simply by the general formula of a weighted mean

$$\tilde{\mu} = \frac{\omega_1}{\omega_1 + \omega_2}\tilde{\mu}_1 + \frac{\omega_2}{\omega_1 + \omega_2}\tilde{\mu}_2.$$

Then the recursive mean on $\mathcal{M}$ is $\hat{\mu}_{a,b}^{(\omega_1+\omega_2)} = \exp_\eta(\tilde{\mu})$.

**Theorem 3.1** (Unbiasedness). *Let $(\sigma, \omega)$ be a measurable space with probability measure $\omega$, and let $x$ denote a measurable function on $\sigma$ taking values in $\mathbb{R}^k$, where $k \in \mathbb{N}$. The expectation of $x$ can be defined as $\mathbb{E}(X) = \int_\sigma x d\omega$. Let $\{x_i\}_{i \in \mathbb{N}}$ be a set of n i.i.d. samples from distribution of $x$, then the recursive mean estimator $\hat{\mu}_{0,1}^{(n)}$ of the samples is an unbiased estimator of $\mathbb{E}(x)$.*

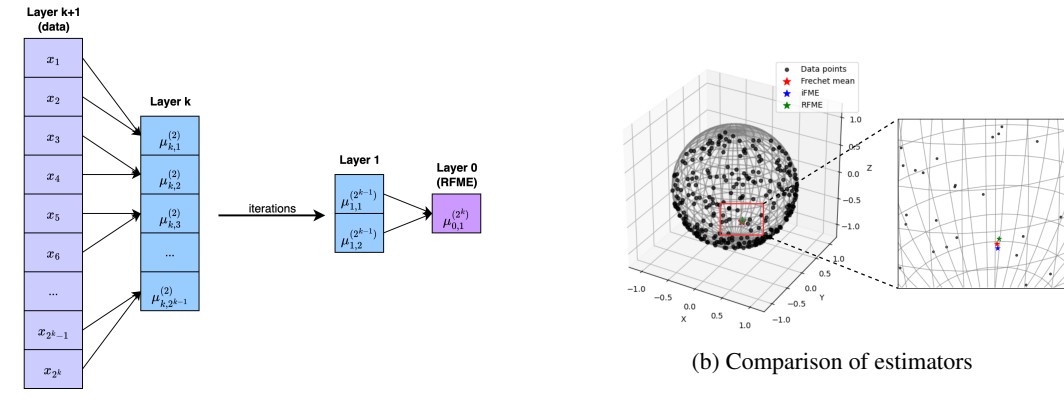

(a) Mean estimation

(b) Comparison of estimators

Figure 1: Left: Illustration of recursive mean estimation with a sample size $2^k$. Right: Comparison of the Fréchet mean via gradient descent method (red), iFME (green) and RFME (blue), using uniformly distributed data on hemisphere.

*Proof.* Let $\tilde{\mu}$ be the recursive mean and $\tilde{\mu}_1$, $\tilde{\mu}_2$ be the left and right children (on the tangent space).

The conclusion is clear for $n = 1$.

If $n = 2$, then recursive mean is the average of two data $\tilde{\mu} = \frac{1}{2}x_1 + \frac{1}{2}x_2$, hence $\mathbb{E}(\tilde{\mu}) = \frac{1}{2}\mathbb{E}(x_1) + \frac{1}{2}\mathbb{E}(x_2) = \mathbb{E}(X)$

If $n = 3$, then $\tilde{\mu} = \frac{2}{3}\tilde{\mu}_1 + \frac{1}{3}\tilde{\mu}_2$, where $\tilde{\mu}_1$ is the mean in $n = 2$ case and $\tilde{\mu}_2 = x_3$, hence $\mathbb{E}(\tilde{\mu}) = \frac{2}{3}\mathbb{E}(X) + \frac{1}{3}\mathbb{E}(x_3) = \mathbb{E}(X)$

By induction assumption, we have $\mathbb{E}(\tilde{\mu}) = \mathbb{E}(X)$ for $n \leq m-1$, then for $n = m$, the binary tree has depth $\lceil \log_2^m \rceil + 1$. $\tilde{\mu}_1$ has weight $\frac{2^{\lceil \log_2^m \rceil - 1}}{m}$, and $\tilde{\mu}_2$ has weight $\frac{m - 2^{\lceil \log_2^m \rceil - 1}}{m}$. Since both of the child come from a subtree, which are cases for $n \leq m-1$, we can conclude that $\mathbb{E}(\tilde{\mu}_1) = \mathbb{E}(\tilde{\mu}_2) = \mathbb{E}(X)$, hence $\mathbb{E}(\tilde{\mu}) = \frac{2^{\lceil \log_2^m \rceil - 1}}{m}\mathbb{E}(\tilde{\mu}_1) + \frac{m - 2^{\lceil \log_2^m \rceil - 1}}{m}\mathbb{E}(\tilde{\mu}_2) = \mathbb{E}(X)$

Therefore, we have $\mathbb{E}(\tilde{\mu}) = \mathbb{E}(X)$ for $n \in \mathbb{N}$. The recursive mean $\tilde{\mu}$ is unbiased on the tangent space $T_\eta \mathcal{M}$, thus $\tilde{\mu}_{0,1} = \exp_\eta(\tilde{\mu})$ is unbiased on the manifold $\mathcal{M}$.

$\square$

**Theorem 3.2** (Consistency). *Let $var(X)$ and $var(\hat{\mu}_{0,1}^{(n)})$ be the variance of distribution of x (defined in 3.1) and the recursive mean estimator, then we have $var(\hat{\mu}_{0,1}^{(n)}) = \frac{1}{n}var(X)$.*

*Proof.* We follow the same idea in the proof of unbiasedness.

If $n = 1$, $var(\tilde{\mu}) = var(X)$.

If $n = 2$, $var(\tilde{\mu}) = var(\frac{1}{2}x_1 + \frac{1}{2}x_2) = \frac{1}{2}var(X)$

If $n = 3$, $var(\tilde{\mu}) = var(\frac{2}{3}\tilde{\mu}_1 + \frac{1}{3}\tilde{\mu}_2) = \frac{4}{9} * \frac{1}{2}var(X) + \frac{1}{9}var(X) = \frac{1}{3}var(X)$

Note that nodes on the same layer come from different data, thus they are independent and covariance is 0.

By induction, assume we have $var(\tilde{\mu}) = \frac{1}{n} var(X)$ for $n \leq m-1$, then for $n = m$,

$$
\begin{aligned}
var(\tilde{\mu}) &= var(\frac{2^{\lceil \log_2^m \rceil - 1}}{m} \tilde{\mu}_1 + \frac{m - 2^{\lceil \log_2^m \rceil - 1}}{m} \tilde{\mu}_2) \\
&= (\frac{2^{\lceil \log_2^m \rceil - 1}}{m})^2 var(\tilde{\mu}_1) + (\frac{m - 2^{\lceil \log_2^m \rceil - 1}}{m})^2 var(\tilde{\mu}_2) \\
&= (\frac{2^{\lceil \log_2^m \rceil - 1}}{m})^2 * \frac{1}{2^{\lceil \log_2^m \rceil - 1}} var(X) + (\frac{m - 2^{\lceil \log_2^m \rceil - 1}}{m})^2 * \frac{1}{m - 2^{\lceil \log_2^m \rceil - 1}} var(X) \\
&= \frac{m}{m^2} var(X) \\
&= \frac{1}{m} var(X)
\end{aligned}
$$

Therefore, $var(\tilde{\mu}) = \frac{1}{n} var(X)$ holds for $n \in \mathbb{N}$. We then have $var(\hat{\mu}_{0,1}^{(n)}) = \frac{1}{n} var(X)$. $\qquad\square$

From theorems 3.1 and 3.2, we conclude that when $n \to \infty$, $Bias(\hat{\mu}_{0,1}^{(n)}) = 0$, and $var(\hat{\mu}_{0,1}^{(n)}) = 0$. This indicates the weak consistency and the recursive mean asymptotically converges to the true mean.

Lastly, we consider the step size.

**Theorem 3.3.** *Given two estimates of $\mu$, $\mu_k$ and $\mu_{k'}$, and geodesic $\alpha_t(\mu_k, \mu_{k'})$, the minimizer of the Fréchet variance is attained at $t = \frac{k'}{k+k'}$.*

*Proof.* Recall that $E(x) = \frac{1}{2} \sum_{i=1}^{n} d^2(x, x_i)$ is variance energy which we aim to minimize. We have that $\mu_k = \operatorname{argmin}_{x \in \mathcal{M}} \frac{1}{k} \sum_{i=1}^{k} d^2(x, x_i)$ and $\mu_{k'} = \operatorname{argmin}_{x \in \mathcal{M}} \frac{1}{k} \sum_{i'=1}^{k'} d^2(x, x_{i'})$. Let $\alpha(t)$ be a geodesic such that $\alpha(0) = \mu_k$ and $\alpha(1) = \mu_{k'}$. We aim to minimize the energy along the geodesic, i.e.,

$$
E(\alpha(t)) = \frac{1}{2(k+k')} \sum_{i=1}^{k+k'} d^2(\alpha(t), x_i) = \frac{1}{2(k+k')} \left[ \sum_{i=1}^{k} d^2(\alpha(t), x_i) + \sum_{i'=1}^{k'} d^2(\alpha(t), x_{i'}) \right].
$$

The gradient with respect to the geodesic is

$$
\nabla_{\alpha(t)} E(\alpha(t)) = -\frac{1}{(k+k')} \left[ \sum_{i=1}^{k} \log_{\alpha(t)} x_i + \sum_{i'=1}^{k'} \log_{\alpha(t)} x_{i'} \right].
$$

Since $\mu_k$ is the minimizer of the first summand and $\mu_{k'}$ is the minimizer of the second summand we thus have $-\frac{1}{(k+k')} \left[ k \log_{\alpha(t)} \mu_k + k' \log_{\alpha(t)} \mu_{k'} \right]$. Evaluating this at $t = 0$ we get $-\frac{k'}{(k+k')} \log_{\mu_k} \mu_{k'}$ as $\log_{\alpha(0)} \mu_k = \log_{\mu_k} \mu_k = 0$ as desired. $\qquad\square$

## 4 EXPERIMENTAL RESULTS

### 4.1 SIMULATION ON SPHERE $S^2$

All experiments in this section were conducted on a 8GM RAM laptop, running Jupyter notebook on Mac with an Apple M1 chip. We evaluate the performance of RFME by comparing with iFME and empirical FM estimator with gradient descent method (eFM). As mentioned earlier, iFME is an add-one-each-time approach and eFM estimator is calculated by minimizing the overall squared distance to all data points.

**Uniform Distribution:** We first employ the uniform spherical sampling method, which, as the name suggests, selects points from a surface of a sphere such that each point has an equal probability of being chosen. To guarantee the uniqueness of the mean, we bound our sample to one hemisphere. The set up of the method includes picking up an azimuthal angle $\phi \sim U(0, \pi)$ (hemisphere) and

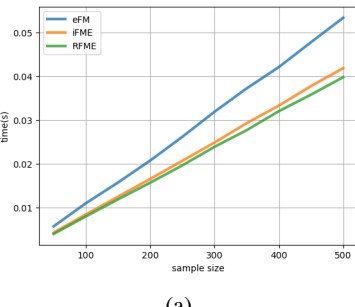

(a)

| | n = 10 | n = 100 | n = 1000 |
|---|---|---|---|
| eFM | $0.0013(4.9 \times 10^{-4})$ | $0.011(2.4 \times 10^{-3})$ | $0.10(2.5 * 10^{-2})$ |
| iFME | $0.0008(1.6 \times 10^{-4})$ | $0.0084(2.4 \times 10^{-4})$ | $0.085(4.3 * 10^{-3})$ |
| RFME | $0.0008(1.4 \times 10^{-4})$ | $0.0080(2.2 \times 10^{-4})$ | $0.078(2.2 * 10^{-3})$ |

(b)

Figure 2: (a) Time comparison among eFM, iFME and RFME. The time is averaged over 200 iterations, with sample sizes ranging from 50 to 500. (b) Stability analysis. The averaged time (standard deviation) is reported over 200 iterations for sample size n=10, 100, 1000. To mitigate the impact of system-related anomalies, 5% of most deviated data are excluded.

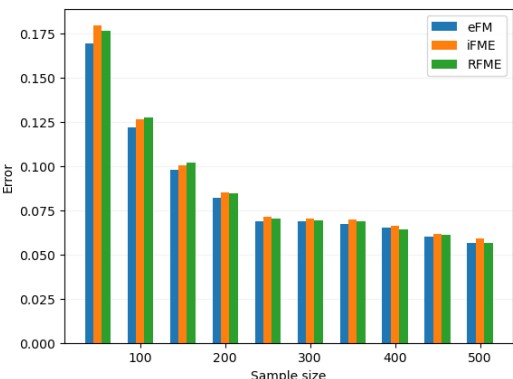

Figure 3: Error comparison among eFM, iFME and RFME, on samples drawn from uniform distribution.

a polar angle $\theta \sim \arccos U(-1, 1)$, then convert into Cartesian coordinates $(x, y, z) = (\sin\theta * \cos\phi, \sin\theta * \sin\phi, \cos\theta)$. This setting enables us to generate data with an expected mean of (0,1,0).

The performance of the eFM, iFME, and RFME is evaluated in terms of time consumption (Fig 2a) and distance from the expected mean (Fig 3). Simulation results show that all three methods have similar accuracy, and their estimated converge towards to the expected value as the data size gets larger. However, in terms of efficiency, RFME performs best, while eFM is the least efficiency. Table 2b presents the stability of the algorithms, showing that eFM takes approximately 1.5 times longer than the other two methods and also has the highest standard deviation. In contrast, RFME demonstrates both the lowest time consumption and the smallest variability.

**Von Mises-Fishers Distribution:** We also want to evaluate the performance of the method under different levels of data dispersion. The von Mises-Fishers distribution, a Gaussian-like probability distribution on the $(p-1)$-sphere $S^{p-1}$ in $\mathbb{R}^p$, has density function $f_p(x; \mu, \kappa) = C_p(\kappa)\exp(\kappa\mu^T x)$, where $C_p(\kappa)$ is the normalizing constant. For $p = 3$, this constant is given by $C_3(\kappa) = \frac{\kappa}{4\pi\sinh(\kappa)}$ (Watson (1982)). Here $\kappa$ is the concentration parameter, with larger values indicating the stronger concentration of data around the $\mu$, the mean direction. We fix the sample size at 500 and repeat the experiment 100 times, applying methods with various value of $\kappa$. As shown in Figure 4a, when data is highly concentrated, the gradient descent algorithm performs best since the centroid of the data can be approached quickly. In contrast, for more dispersed data, iFME and RFME outperform eFM, as their performance only depends on the number of the points. Compared to iFME, RFME is more efficient due to its lower iteration count. Additionally, the von Mises-Fishers distribution does not constrain the samples to lie within a hemisphere, it is expected to see the estimation accuracy of methods degrades when the data dispersion increases, as we can see from Figure 4b.

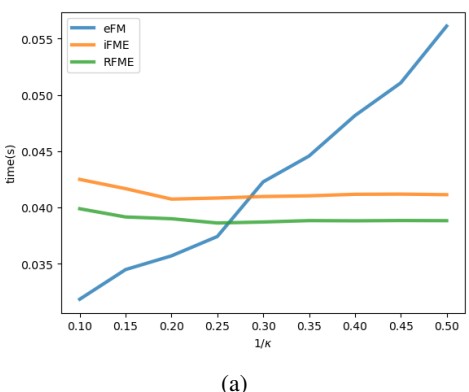
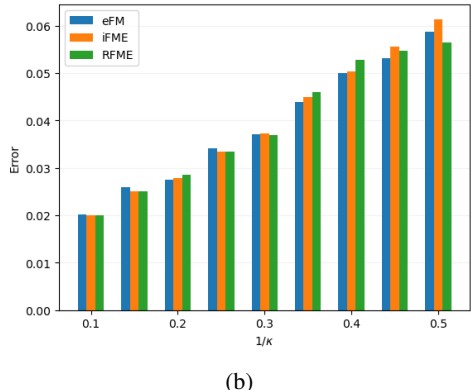

(a)                                                    (b)

Figure 4: Time and error comparisons of eFM, iFME, and RFME. The x-axis represents $1/\kappa$, where the rightmost value corresponds to the most deviated dataset.

## 4.2 REAL CASES

In statistical shape analysis, the geometric shape of objects is typically studied independently of its location, size and orientation. This motivates the idea of defining the group action on the manifold. Given a Lie group $G$, the group action $G * \mathcal{M} \to \mathcal{M}$ is defined such that it satisfies both the identity and compatibility properties. Under this action, each member of the group corresponds to a smooth transformation of the manifold. For an element $p \in \mathcal{M}$, the orbit of $p$ under a group $G$, denoted $[p]$, is defined as $[p] = \{g * p : g \in G\}$, and the quotient space $\mathcal{M}/G$ is the collection of all such orbits on the manifold $\mathcal{M}/G = \{[p] : p \in \mathcal{M}\}$.

Using the translation group $\mathbb{R}^n$, scaling group $\mathbb{R}^+$, and rotation group $SO(n)$, Kendall (1984) introduced the Kendall's shape space as a quotient space

$$\mathcal{L}_{n,k}/(\mathbb{R}^n \rtimes (\mathbb{R}^+ \times SO(n)),$$

where $\mathcal{L}_{n,k} = \{X \in \mathbb{R}^{n \times k} | \dim(\mathrm{span}(X)) = n\}$ and $\rtimes$ is the semi-product of groups. In this formulation, the objects geometric shape corresponds to a point in the shape space.

Let $X \in \mathcal{L}_{n \times k}$ be an ordered k-tuple representing the contour of an object, equipped with Euclidean metric. After centering and normalizing, we get an representative $\hat{X}$ of its orbit $[\hat{X}]$ under $SO(n)$, where $\hat{X} = \frac{X - v 1_k^T}{\|X - v 1_k^T\|}$ and $v$ is the column-wise mean. The orbit lies on a unit hypersphere $\mathbb{R}^{(n-1)k-1}$, so that we can apply the spherical metric, e.g. the log map is given by

$$\log_{[\hat{X}_1]}([\hat{X}_2]) = \frac{\theta}{\sin(\theta)}(O^* \hat{X}_2 - \cos(\theta)\hat{X}_1),$$

with $\theta = \arccos(\langle \hat{X}_1, O^* \hat{X}_2 \rangle)$, and $O^* = \mathrm{argmax}_{O \in SO(n)} \langle \hat{X}_1, O\hat{X}_2 \rangle$ is the optimal rotation aligning $\hat{X}_2$ to $\hat{X}_1$.

In this section, we use two data set to evaluate the models. The first one is the hammer contour data, extracted from MPEG-7 image dataset (Thoum et al. (2008), Thourn & Kitjaidure (2009)). The MPEG-7 dataset consists of 70 types of objects with 20 different shapes, and is widely used in area of computer vision. Each contour in the dataset is represented by a ordered set of 100 landmarks with dimension 2.

The second contains the corpus callosum shapes from the ADNI (Alzheimer's Disease Neuroimaging Initiative) database (Mueller et al. (2005)). The corpus callosum is a wide, thick bundle of nerve fibers that connects the two hemispheres of the brain. It helps the transmission of neural signals between the two sides and plays a key role in coordinating sensory perception, movement, and cognitive functions of people. The samples are extracted from the magnetic resonance imaging(MRI) scans, and also represented by an ordered set of 51 landmarks in a 2D plane.

Fig 5 presents some sample data, along with the estimators of Fréchet mean. we can find from the plots that the shaded, dashed and dotted estimates are nearly identical, with only minor differences

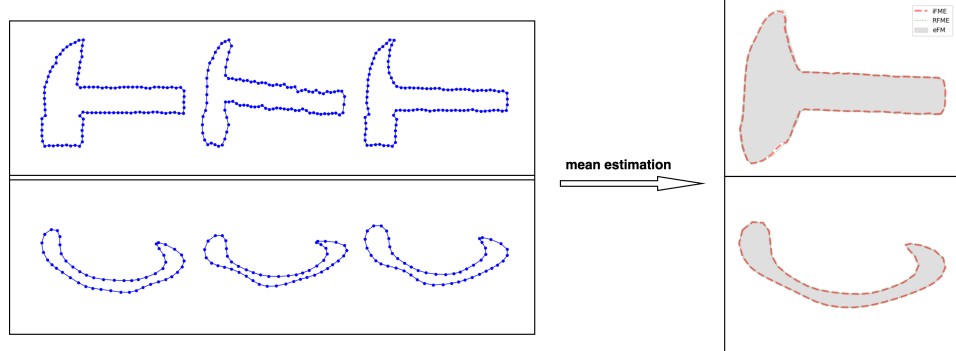

Figure 5: Three hammer samples (out of 9) and corpus callosum samples (out of 409), and mean estiations from eFM (shaded), iFME (dashed line) and RFME(dotted line).

near the head region of the hammer data. To assess the accuracy, we adopt the sum-of-squared error (SSE), as the truth Fréchet mean corresponds to the minimum SSE. For the hammer data, eFM has an SSE of $4.58 \times 10^{-3}$, while the iFME and RFME has SSE of $4.54 \times 10^{-3}$. In the case of corpus callosum data, eFM results in a SEE of $2.78 \times 10^{-3}$, while iFME and RFME achieve SSE of $2.77 \times 10^{-3}$. Compared to iFME, the RFME has slightly lower SSE, though the difference is negligible. For eFM, the gradient stopping criterion is set to be $1 \times 10^{-4}$; using a smaller threshold could yield more accurate estimation but at the cost of increased computational time. Overall, both visual and numerical results indicate that RFME is an effective and efficient alternative to the traditional gradient descent methods in many situations.

## 5 CONCLUSION

This paper introduces a Recursive Fréchet Mean Estimator (RFME), a tree-based inductive algorithm for approximating the barycenter of data on the hypersphere. We prove the RFME is unbiased and converges asymptotically to the true Fréchet mean. Empirical results from simulations and real-world case studies show that RFME is significantly more efficient than the traditional gradient descent approach. Conceptually, RFME is a generalization of iFME. While iFME can be seen as a tree with only two nodes on each layer: one is mean estimator and the other is a data point; RFME has more flexible structure. Although it requires more storage, it supports further acceleration through advanced techniques such as parallel computing, and offering more potential in both theory and application.

In modern data analysis and machine learning, online learning has become increasingly important, as data often come in as a stream. This is because either the full dataset is too large to store or use at once or the data are generated continuously over time. This calls for adaptive systems that can update incrementally without revisiting the entire dataset. RFME fits into this setting well: it enables the current estimator to be updated with incoming data only, thus reducing time, computational resources, and storage requirements.

In addition, while this paper only focus on applying RFME on the hypersphere and Kendall's shape space, which has constant positive curvature, the estimator itself relies solely on the geodesic. As such, it can be readily generalized to a boarder class of Riemannian manifolds. Future work may explore extending the method to general positive curved spaces, which has more complex structure than NPC spaces, as well as on analyzing the theoretical properties such as the error bound for finite data sets.

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

## A  Appendix

### A.1  The Use of Large Language Models (LLM)

We use a Large Language Model (ChatGPT) as an assistant to help rephrase and improve the clarity of wording. The LLM only serves as a tool to enhance the expression while ensuring the original meaning is preserved.

