# OpenReview forum: "Efficient Recursive Fr\'echet Mean Estimation"
_ICLR.cc/2026/Conference — ICLR 2026 Conference Withdrawn Submission_

### Official Review · Reviewer_Riww · 2025-10-26

**Soundness:** 1
**Presentation:** 2
**Contribution:** 1
**Rating:** 2
**Confidence:** 5

**Summary:**

The paper proposes the Recursive Mean algorithm, which is a variant of the inductive mean estimator proposed in [1]. The suggested algorithm uses a binary tree structure, which can take advantage of parallel programming unlike the inductive mean. The authors claim weak convergence guarantees (Theorems 3.1 and 3.2) and provide numerical experiments (Section 4).

[1] Karl Theodor Sturm. Probability measures on metric spaces of nonpositive curvature. Heat Kernels and Analysis on Manifolds, 2000.

**Strengths:**

* The binary tree–based approach seems new and worth exploring, particularly given the fact that this tree-based approach is favorable for parallel computation.

* The authors provided experiments under various settings.

* The background section is well written; it is concise while including prior works and necessary information.

**Weaknesses:**

* Some notations are causing confusion. For instance, in Line 160, the authors used $x_i$ as the points on the manifold. However, in Theorem 3.1, the authors used $x_i$’s as the coordinate chart. Also, $\eta$ denotes the step size in Line 118, but later in Line 204, $\eta$ is any point on $M$.

* The conditions on a manifold for the theoretical results to hold are not precisely stated (related to the next bullet point).

* I believe Theorem 3.1 is both unclear and wrong in several ways.
    * First of all, based on the theorem statement, $E(x)$ is an element in $\mathbb{R}^k$, and $\hat{\mu}_{0,1}^{n}$ is an element in $M$. The meaning of unbiasedness does not make sense between elements in different spaces.
    * Looking at the proof, I thought $x$ should be a coordinate chart at $\eta$, if I understood correctly. Then, $E(x)$ is an extrinsic mean, which has no direct relationship with the population Frechet mean (which is intrinsic). Unless the paper’s main agenda is the extrinsic mean (which I believe is not), the theorem statement (unbiasedness to extrinsic mean) does not correspond to the main message of the paper (unbiasedness to intrinsic mean).
    * Or, if $x$ is not a coordinate chart and the authors intended $x$ as the points on the manifold (with the addition operator meaning geodesic interpolation as in [1]), then the proof is not valid; there is no linearity of expectations between geodesic interpolations. One should approach this as in [1, Theorem 4.7] and invoke inequalities and curvature arguments.
    * What I think is that the authors intended small $x$ as the data point and large $X$ as the coordinate chart at $\eta$, defined by $X = \log_\eta x$. Then one substitutes $x$ in the proof by $X$ (am I correct?). If this is the case, the proof is unfortunately not valid. In this case, the authors’ claim can be summarized by $E(\tilde{\mu}) = E(X)$ implies $E(\hat{\mu}) = E(\exp_\eta \tilde{\mu}) = E(\exp_\eta X) = E(x)$. However, the second equality is false unless $\exp_\eta(\cdot)$ is affine (which is an extremely rare case). Counterexample: even in simple $\mathbb{R}$, if $\tilde{\mu} \sim N(0,1)$, $X \sim N(0,4)$, and $\exp_\eta(v) = v^2$, then $E(\tilde \mu) = 0 = E(X)$, while $E(\exp_\eta \tilde{\mu}) = 1 \neq E(\exp_\eta X) = 4$.
    * Even if we assume the above issues are resolved, the proof is still not valid. There should be an assumption that the Riemannian logarithmic map is well-defined at $\eta$, which is guaranteed only if the estimators lie inside the injectivity radius of $\eta$ (which should be a concern as the main application here is the sphere). Particularly, such an assumption is not straightforward, as $\hat{\mu}$ is a random quantity.

* Theorem 3.2 is also invalid, due to the failure of Theorem 3.1. First of all, again it is not clear whether $x$ is a coordinate chart or a point on the manifold. If $X$ is not a coordinate chart, then the proof is not valid (the summation formula of variance is not true). If $X$ is the coordinate chart at $\eta$, then the proof is in fact valid if $\eta = \mu$, $E(\log_\mu \hat{\mu}) = 0$ (true if unbiased), and all the logarithmic maps are well defined: $\mathrm{Var}(\hat{\mu}) = \int \|\log_{\mu} \hat{\mu}\|^2 d\omega = \mathrm{Var}(\tilde{\mu}) = \mathrm{Var}(X)/n$. However, as mentioned above, the unbiasedness proof is invalid, so even when $\eta = \mu$, the proof is not valid unless we have unbiasedness.

* While the authors claim computational advantages, it seems like they did not provide computational complexity. In fact, I do not find the proposed algorithm computationally efficient (related to the question part).

**Questions:**

* Minor: Line 107 – use brackets to express the mean.

* Minor: Line 112 – should be $\in$ instead of =.

* It is well known that the inductive mean is *not* invariant under permutations of the data [1]. I expect similar behavior for this estimator. Is that true?

* At a glance, assuming there is an oracle that computes the geodesic interpolation in $O(1)$, it seems that the computational complexity of the estimator is $\Theta(n)$, am I correct? Naively, when $n = 2^k$ for some $k \in \mathbb{N}$, the computational complexity seems to be $n/2 + n/4 + \dots + 1 = \Theta(n)$. This should be the same computational complexity as the inductive mean estimator. Of course there is a gain from parallel computation, but in that case one should specify what one meant by computational advantage.

---

### Official Review · Reviewer_xQ4T · 2025-11-03

**Soundness:** 1
**Presentation:** 2
**Contribution:** 1
**Rating:** 0
**Confidence:** 2

**Summary:**

Paper presents a recursive generalization to the natural incremental estimator for the Frechet mean of a data set on manifold (the Frechet mean is a generalization of the mean to manifolds and should be thought of as the least squared distance solution point with respect to the input data). The paper show the consistency of the estimator and shows empirical performance in some applications

**Strengths:**

The problem considered could be of practical importance and the present algorithm has improved parallel efficiency compared to the previosu algorithms

**Weaknesses:**

I am actually not convinced that the consistency proof presented in the paper checks out. In fact, I am not sure how I should interpret the mean in general manifold settings. Furhter even given this, the theoretical contribution seems minor and as thus the empirical contribution.

**Questions:**

-> I am not entirely sure I understand the setting of theorem 1. x_i are points on a manifold right? How should I interpret EX? The frechet mean is the natural interpretation of this. But the paper seems to consider intergration over R^d (which I guess the manifold is embedded in) in which case the two notions don't agree. One can then think maybe that the mean should be interpreted as some push forward on the space. Then one needs to show that the Frechet mean does indeed satisfy this consistency which is some version of the "unbaisedness" of the exponential map. But this is not obvious to me neither is it discussed in the paper

---

### Official Review · Reviewer_13Mv · 2025-11-04

**Soundness:** 2
**Presentation:** 3
**Contribution:** 2
**Rating:** 2
**Confidence:** 3

**Summary:**

This paper proposes a tree-based recursive algorithm to calculate the Frechet mean for data in a Riemannian manifold. Basic consistency of the proposed estimator is studied, and the numerical performance of the proposed methods is compared with two existing methods in the sphere cases $\mathbb{S}^2$.

**Strengths:**

Non-Euclidean data have become increasingly common in scientific research, and the development of related statistical tools and methodologies has become an important line of work in this area. This paper proposes a new algorithm for calculating the Fréchet mean, which shows superior numerical performance compared to existing methods in both computing time and accuracy.

This paper is well-organized and clearly written.

**Weaknesses:**

1. In the numerical studies, the authors consider only one Riemannian manifold that is the sphere, which is not very comprehensive.
2. The numerical results are only slightly better than existing methods, making the contribution of this paper less pronounced.
3. Some of the theoretical results are not clear.
4. The basic consistency results are not sufficient, and the detailed theoretical properties of the proposed algorithm need to be further studied.

**Questions:**

1. What spaces can the proposed algorithm be applied to? Riemannian manifolds, geodesic spaces, or general metric spaces?
2. In Theorem 3.1, what properties should the space $\sigma$ have? Should it be a Euclidean space or more general spaces like metric spaces or geodesic spaces? If it is the latter case, how is the integral $E(X)$ defined?
3. Similarly, when $X$ is not a scalar, how is its variance defined in Theorem 3.2?
4. Could the authors provide an anonymous GitHub repository to share their code?

---

### Author Response · Authors · 2025-11-17

We thank all the reviewers for their hard work and elaborate feedback. We will work on the suggested feedback. As the scores are lower than "borderline" we will withdraw and resubmit once we have incorporated the suggested improvements.

Thank you.

---

### Note · Authors · 2025-11-17

I have read and agree with the venue's withdrawal policy on behalf of myself and my co-authors.